

# Mass harvested per trunkload as a constraint to forage consumption by the African savanna elephant (*Loxodonta africana*)

Bruce W. Clegg[1] and Timothy G. O'Connor[2]

[1] The Malilangwe Trust, Chiredzi, Zimbabwe
[2] School of Animal, Plant, and Environmental Sciences, University of Witwatersrand, Johannesburg, Gauteng, South Africa

## ABSTRACT

**Background**. African elephants can convert woodland to shrubland or grassland. Moderate conversion observed at low elephant densities may improve conditions for other animals, while extensive transformation at high densities may reduce plant and animal diversity. The threshold density separating facilitation from habitat destruction varies spatially and is partly determined by food choice, which differs between adult bulls and members of breeding herds. When elephants consume herbaceous forage, woodland damage is low but this increases when woody plants are the primary food source. Consequently, an understanding of diet selection by elephants is important for forecasting the degree of vegetation conversion. One hypothesis is that elephants select forage that provides the highest rate of intake. The mass harvested per trunkload is a constraint to intake and therefore this study sought to determine if trunkload mass changes seasonally; varies across common forage types utilised by elephants; and differs between adult bulls and members of breeding herds.

**Methods**. Mechanistic models were used to estimate the mass harvested per trunkload of green grass, mixed green and dry grass, forbs, and leaves and bark from woody plants across a heterogenous, semi-arid savanna at a daily time step for one annual cycle. Separate models were constructed for adult bulls and members of breeding herds.

**Results**. Harvestable mass changed seasonally for herbaceous forage and for leaves from woody plants but was constant for canopy bark. The maximum average trunkload mass of green grass was >75 times heavier than the bite mass reported for other grazers while trunkloads of leaves from woody plants were only eight times heavier than the bite mass reported for other browsers. This is attributed to the advantage provided by the trunk, which increases harvestable mass beyond the constraint of mouth volume, particularly when feeding on grass. Herbaceous forage yielded heavier trunkloads than leaves and bark from woody plants during the wet season, but this was reversed in the dry season. Adult bulls harvested heavier trunkloads than members of breeding herds for all forage types except forbs; and adult bulls harvested disproportionately large trunkloads of grass and bark.

**Conclusion**. The strong correlation between the model outputs and well-established trends in the seasonal changes in elephants' diet suggests that elephants are preferential foragers of the largest trunkload on offer. Consequently, they are grazers when suitable herbaceous forage is available, and browsers when it is scarce. Green grass provides

Corresponding author
Bruce W. Clegg,
bruce@malilangwe.org

adult bulls with disproportionately large trunkloads and, therefore, adult bulls are predicted to have a strong preference for green grass. Availability of suitable green grass during the dry season may therefore buffer woodlands from heavy impact by adult bulls. Consequently, where possible, protected areas with elephants should aim to include key grass resources.

# INTRODUCTION

African savanna elephants can significantly transform their habitats. Since the 1960s, conversion of forest and woodland to shrubland or grassland has been a characteristic feature of reserves with high densities of elephants in east, central and southern Africa (*Buechner & Dawkins, 1961*; *Lamprey et al., 1967*; *Field, 1971*; *Anderson & Walker, 1974*; *Guy, 1981*; *Dublin, Sinclair & McGlade, 1990*; *Trollope et al., 1998*; *Mapaure & Campbell, 2002*; *Guldemond & Van Aarde, 2008*; *Valeix et al., 2011*; *Coetsee et al., 2023*). This has been brought about by elephants felling or ring barking mature trees and suppressing recruitment from the shrub layer (*Western & Maitumo, 2004*), although factors such as fire, drought and pressure from other herbivores have also played a role (*Barnes, 1983*; *Pellew, 1983*; *Dublin, Sinclair & McGlade, 1990*; *Birkett, 2002*; *Coetsee et al., 2023*). At low to medium elephant densities the modification of woody vegetation may improve conditions for other animals (*e.g.,* *Parker, 1983*; *Valeix et al., 2011*; *Landman & Kerley, 2014*), but at high densities the transformation is potentially damaging, leading to concern over the possible extirpation of plant species and the loss of animal species whose survival is dependent on forest or woodland habitat (*Laws, 1970*; *Herremans, 1995*; *Cumming et al., 1997*; *Fenton et al., 1998*; *Lombard et al., 2001*; *Fritz et al., 2002*; *Braswell & Slusar, 2005*; *Botes, McGeoch & Van Rensburg, 2006*; *Kerley & Landman, 2006*; *O'Connor, Goodman & Clegg, 2007*; *Landman et al., 2014*). The threshold elephant density that separates facilitation from habitat destruction varies across environments (*Guldemond & Van Aarde, 2008*) and is partly determined by what elephants choose to eat (*O'Connor, Goodman & Clegg, 2007*).

Savanna elephants feed from both herbaceous and woody vegetation. Green grasses and forbs are consumed in large quantities when they are available, with diet switching to leaves, bark, and roots of woody plants when herbaceous forage is scarce (*Buss, 1961*; *Guy, 1976*; *Cerling et al., 2009*; *Codron et al., 2011*; *Shannon, Mackey & Slotow, 2013*; *Gill et al., 2023*). The methods elephants use to harvest food from shrubs and trees are potentially destructive, particularly when extracting bark and roots (*Clegg & O'Connor, 2016*), and therefore greater transformation of woody vegetation occurs when a large proportion of elephants' diet is from woody plants (*O'Connor, Goodman & Clegg, 2007*). In contrast, when diet is predominantly herbaceous forage, trees and shrubs are less impacted. The age and sex structure of an elephant population may also influence the degree of habitat transformation because adult bulls are twice the body size of cows (*Owen-Smith, 1988*) and are therefore stronger and more capable of impacting larger trees (*Guy, 1976*; *Hiscocks,*

*1999*). An understanding of the drivers of diet composition of both adult bulls and members of breeding herds (adult cows and subadults of both sexes) is therefore key to forecasting the degree of woody vegetation conversion.

Optimal foraging theory suggests that foraging behaviour and fitness are linked by a particular currency (*Pyke, 1984*). If elephants rank potential food items according to their contribution to overall fitness, and use these rankings to select a diet that best meets their needs, what currency do they use? To answer this question, it is necessary to identify the components of forage that are attractive to elephants. Plant cells can be divided into cell solubles contained in the cytoplasm and cell wall material that forms a structural box around the cytoplasm (*Lyons, Machen & Forbes, 1996*). Cell solubles, which include protein, sugars, starch, and lipids are rapidly and almost completely digestible. In comparison, the cell wall contains slowly digestible fibre, which includes hemicellulose, cellulose and the mostly indigestible lignin (*Lyons, Machen & Forbes, 1996*; *Buxton & Redfearn, 1997*; *Shipley, 1999*). Elephants are hindgut fermenters (*Van Hoven, Prins & Lankhorst, 1981*) and unlike ruminants their digestive system does not have blocking structures that limit the rate of passage of material through the gut, and therefore they experience short ingesta retention times. Retention time in elephants may be as short as 14 h (*Eltringham, 1982*) compared to 70–100 h in cattle (*Owen-Smith, 1988*). Because food is retained in the gut for a short time, fermentation of slowly digesting cell walls is limited (*Van Hoven, Prins & Lankhorst, 1981*; *Meissner et al., 1990*), and therefore elephants should be more reliant on forage that is rich in rapidly digestible cell solubles than animals that obtain energy and nutrients from digesting cell walls more completely. Given the above, it has been hypothesised that elephants seek to maximise their rate of intake of plant cell contents, and that they rank potential food items by using intake rate as a scaling currency (*O'Connor, Goodman & Clegg, 2007*).

If elephants rank the profitability of forage according to how rapidly cell contents can be consumed, what are the factors that influence this? The rate at which mammalian herbivores consume forage is influenced by plant secondary metabolites that chemically defend against herbivory (*Freeland & Janzen, 1974*; *Iason, 2007*), and by constraints to the physical mechanics of the foraging process (*Spalinger & Hobbs, 1992*; *Gross et al., 1993*; *Laca, Ungar & Demment, 1994*; *Shipley et al., 1994*; *Hobbs et al., 2003*; *Shipley, 2007*). Potential physical constraints to an elephant's foraging include: (1) the time to locate a suitable patch of forage, which is determined by the density of forage patches in the landscape; (2) the time to harvest and chew a trunkload of food; (3) the mass of forage gathered with each trunkload; (4) the number of trunkloads harvested per forage patch; and (5) the density of cell solubles in the plant tissue (*Clegg, 2010*). These vary across forage types, and across space and time, creating a constantly changing set of potential food choices (*Clegg, 2010*; *Clegg & O'Connor, 2016*; *Clegg & O'Connor, 2017*).

The effect of chemical defence (*Schmitt, Ward & Shrader, 2016*; *Schmitt et al., 2020*), seasonal variation in the density of forage patches (*Clegg & O'Connor, 2017*), and the time to harvest and chew different forage types (*Clegg & O'Connor, 2016*) have been investigated for elephants, but little is known about how trunkload mass varies across forage types, and how this might affect diet. The aim of this study was to understand how variation in

trunkload mass might influence food intake rate. An empirically-based, spatially explicit modelling approach was used. Specifically, we modelled the seasonal changes in the mass harvested per trunkload for the main forage types used by elephants across a spatially heterogenous savanna landscape. Separate models were constructed for adult bulls and members of breeding herds. The aim was to understand how variation in trunkload mass might influence food intake rate, and to parameterize the trunkload mass component of an integrated intake rate model for elephants (*Clegg, 2010*). Specific questions were (1) does the mass harvested per trunkload differ across forage types, (2) does it change seasonally, and (3) does it differ between adult bulls and members of breeding herds? The construction of a model that incorporates all potential constraints to intake rate is a future objective. This paper represents one of the steps towards achieving that goal.

We hypothesised that harvestable mass is influenced by the mass of individual leaves or leaflets when harvesting leaves from woody plants; the mass of the whole plant when eating forbs; the basal area of a tuft and height of the tallest leaf when harvesting grasses; and the diameter and length of a branch when chewing bark off branches of shrubs and trees. Consequently, we first empirically parameterized the relationship between these plant attributes and trunkload mass for each forage type, and then used the derived relationships to model harvestable mass for each forage type over the study landscape at a daily time step.

Portions of this text were previously published as part of a preprint (https://wiredspace. wits.ac.za/items/076c0726-ba2d-4b77-8b5e-0147fa6279f6).

## MATERIALS & METHODS

The study was conducted between November 2001 and July 2003, in the semi-arid savanna of Malilangwe Wildlife Reserve, a fenced protected area of 465 km$^2$, in south-eastern Zimbabwe (20°58′–21°15′S, 31°47′–32°01′E). Permission to conduct the study was granted by The Malilangwe Trust. The reserve has a hot wet season from November to March, a cool dry season from April to August, and a hot dry season from September to October. Mean annual rainfall is 564 mm ($n = 72$; CV = 33.8%), with approximately 84% falling in the hot wet season. Rainfall during the annual cycle under study (2002–2003 rainfall season) was 716 mm. Due to the proximity of the study area to the Indian Ocean ($\approx$200 km), frontal rainfall during the dry season is not uncommon (19.8 mm, 35.1 mm, and 42.4 mm were recorded in July, September and October of 2002, respectively). The average minimum and maximum monthly temperatures range from 13.4 °C (July) to 23.7 °C (December), and 23.2 °C (June) to 33.9 °C (November), respectively (*Clegg, 2010*). Frost is rare. Thirty-eight vegetation types, from open grassland to dry deciduous forest, have been identified on seven geological types, with soils ranging from 90% sand to 41% clay (*Clegg & O'Connor, 2012*). Fire has been used as a tool for rangeland management since 1994. Elephants occurred at a density of 0.3 km$^{-2}$ at the time of study.

This study focused on grass, forbs, and leaves and bark from woody plants because these forage types constitute the bulk of an elephant's diet. Landscape-scale estimation of the mass harvested per trunkload requires knowledge of the availability of each food

type across both space and time. Foraging ultimately involves choices between individual plants and therefore available food resources were described at the individual plant level. We used mechanistic models to estimate the mass harvested per trunkload of each forage type across the study landscape at a daily time step for one annual cycle. The models accounted for variation in botanical composition, seasonal changes in plant phenology, and the foraging height limits of adult bulls and cows. The models were run in a geographic information system that used a spatial framework with 65 landscape units as a base layer. The framework was created by combining a fine-scale vegetation map with 38 vegetation types (*Clegg & O'Connor, 2012*) with maps of areas burnt during the previous year and areas that experienced early woody leaf flush (identified from an October 2002 landsat 7 ETM+ image using a Normalized Vegetation Index). Within each land unit, trunkload mass for adult bulls and members of breeding herds was estimated at a daily time step for each forage type using the following methods. Trunkload mass was estimated using data collected by unobtrusive observation of foraging elephants. Consequently, elephants were not disturbed in any way during data collection.

## Estimation of trunkload mass for grass

When feeding on grass elephants generally uproot and consume a whole tuft with each trunkload. However, for robust perennials only the upper parts of a tuft are eaten, the roots and bases of the tillers remaining unharvested or being discarded. Consequently, for robust perennial grasses the mass of a trunkload was represented by the average mass of the upper portions of the tuft and for other species by the average mass of the whole plant (*Clegg, 2010*).

Elephants avoid eating senescent forage, so a grass tuft is selected if it offers a sufficient ratio of green to dry material. It takes twice as long for elephants to harvest, clean by shaking, chew and ingest a trunkload of mixed green and dry grass compared to one of green grass only (*Clegg & O'Connor, 2016*). Consequently, separate estimates were derived for trunkloads of green grass, and mixed green and dry grass.

The mass of a trunkload when feeding on green grass, $S_{grass\ green}$, varies both spatially and temporally because of differences between grass species, soil types and stages of growth. To capture this variation, $S_{grass\ green}$ was estimated daily for each landscape unit as:

$$S_{grass\ green} = aH_{green}B,$$

where $H_{green}$ is the average height (cm) of the tallest green grass leaf, $B$ is the average basal area (cm$^2$) of a tuft, and $a$ is a constant. Trunkload mass when feeding from mixed grass was estimated in the same way except values for $H_{green}$ were substituted by values for mixed green and dry grass patches.

An estimate of $a$ was derived for each vegetation type using the following approach. A list of grass species was compiled by recording the dominant perennial grass species in each vegetation type (*Clegg & O'Connor, 2012*). Twenty-five tufts, covering a range of leaf heights, basal areas, and vegetation types, were selected for each species on the list. For each tuft, the height above the ground of the tallest leaf and the circumference of the base was measured to the nearest centimetre. The tuft was then uprooted, dried to constant
mass after removing the soil from the roots, and weighed to the nearest gram. The data for individual grass species were then used to construct data sets for each vegetation type by pooling the data for each of the dominant grass species found growing in a vegetation type. Estimates of $a$ were derived for each vegetation type by fitting the above equation to the pooled data sets using the linear regression module of Systat 9 (*SPSS, 1998*).

To derive daily estimates of $H_{green}$ and $H_{mixed}$, each landscape unit was sampled between November 2001 and July 2003 at approximately 3-month intervals. On each occasion, five sample points were positioned in each landscape unit using a stratified random sampling strategy and located in the field using a GPS. At each sampling point, the height of the tallest green and dry grass leaf was measured to the nearest centimetre in $25 \times 1$ m$^2$ quadrats positioned along a 50 m tape. A time series for each landscape unit was constructed by plotting the average height data ($H_{green}$ and $H_{dry}$) against time and interpolating between sample points using the smoothing spline regression module of Kyplot 4.0 (KyensLab Inc., 2002). Data for $H_{mixed}$ were calculated as the average of $H_{green}$ and $H_{dry}$ for each sample point.

The average basal area of tufts in each landscape unit was estimated by randomly selecting between 50 and 100 tufts of each of the dominant perennial grass species and measuring the circumference of each tuft (nearest centimetre). The point at which the coefficient of variation stabilised determined the number of tufts sampled for each species. There is a limit to the size of tufts elephants will uproot in a single trunkload. To estimate the preferred maximum circumference of tufts for adult bulls and members of breeding herds, 100 tufts of *Setaria incrassata* that had been uprooted, fed on and discarded were collected from beds of *Setaria incrassata* where adult bulls only and members of breeding herds only had been feeding. *Setaria incrassata* was chosen as the target species because, of all the grass species at Malilangwe, it has individuals with the largest tuft circumference and therefore provides an opportunity to estimate the upper limit of tuft size selection. The mean tuft circumferences selected by adult bulls, members of breeding herds, and within the plant community as a whole were calculated, and differences between means were tested using a Kruskal–Wallis test and a Dunn *post hoc* test (chosen because data were non-normal and heteroscedastic). The analysis was conducted using R version 4.1.2 (*R Core Team, 2021*), with the 'FSA: Simple Fisheries Stock Assessment Methods' package (version 0.9.5; *Ogle et al., 2023*) being used for the Dunn test. The mean values were used to adjust the tuft circumference data sets for each grass species by setting a ceiling on the size of tufts selected by adult bulls and members of breeding herds. The adjusted tuft circumference data were then used to calculate the average basal area for each species. The average basal area of a tuft selected by adult bulls, $B_{bull}$, was estimated for each landscape unit as:

$$B_{bull} = \sum_{i=1}^{n} B_{i,bull} C_i,$$

where $B_{i,bull}$ is the average basal area of a tuft of the $i$th species selected by adult bulls and $C_i$ is the proportional aerial cover of the $i$th species (refer to *Clegg & O'Connor, 2012* for how the aerial cover of grass species was estimated). The average basal area of a tuft selected

by members of breeding herds, $B_{cow}$, was calculated in a similar way except that $B_{i,bull}$ was replaced by $B_{i,cow}$.

The herbaceous layer of some vegetation types was entirely composed of annual grasses and because elephants also utilise annual grasses the most abundant of these were selected for measurement. However, the seasonal variation in mass of annual grass plants was ignored on account of their rapid natural attrition following cessation of growth. Consequently, only the dry mass of 25 randomly selected plants was measured, from which the average dry mass of a plant was calculated. For these vegetation types, $S_{grass\ green}$ was estimated as:

$$S_{grass\ green} = \sum_{i=1}^{n} W_i C_i,$$

where $W_i$ is the average dry mass of the $i$th annual grass species, and $C_i$ is the relative per cent aerial cover of the $i$th annual grass species.

## Estimation of trunkload mass for forbs

When feeding on forbs (defined in this study as herbaceous dicotyledonous vascular plants), elephants uproot and consume a whole plant with each trunkload. It was assumed that the mass of individual forb plants was greatest at the end of the growing season and declined as the dry season progressed due to browsing, trampling and senescence. It was assumed that the decline in mass was proportional to the change in average forb height and, consequently, trunkload mass when feeding on forbs, $S_{forb}$, was calculated, for each landscape unit, as:

$$S_{forb} = \sum_{i=1}^{n} W_i H_{forb} C_i,$$

where $W_i$ is the average dry mass of the $i$th forb species at the end of the growing season, $H_{forb}$ is the average height of green forbs relative to the maximum average height measured for the landscape unit, and $C_i$ is the proportional aerial cover of the $i$th forb species (refer to *Clegg & O'Connor, 2012* for how the aerial cover of forb species was estimated). $H_{forb}$ was estimated daily for each landscape unit by measuring the height of the tallest forb in the quadrats used to measure grass height and interpolating between average height estimates using the method employed for grass.

In contrast to the pattern of abundance of grasses, the forb component of the herbaceous layer was seldom dominated by a few species. Instead, many forb species occurred in low abundance. Consequently, to reduce data collection to manageable proportions, the average dry mass of each forb species, $W_i$, was estimated, at the end of the growing season, from the dry mass of five randomly selected individual plants.

## Estimation of trunkload mass for leaves from woody plants

The mass of an individual leaf or leaflet influences the mass of leaves gathered with each trunkload from woody plants. When feeding from plants with small leaves, elephants harvest small trunkloads because the mass of individual leaves is low and because small leaves are difficult to handle and many fall from the trunk's grasp. The converse is true for

woody plants with large leaves. To account for variation between species of woody plants, the mass of a trunkload of leaves, $S_{leaf}$, was estimated for each woody species using a linear regression model with the mass of a leaf unit as the predictor variable:

$$S_{leaf,i} = aM_{leaf,i}$$

where $S_{leaf,i}$ is the dry mass of a trunkload of the $i$th woody species, $M_{leaf,i}$ is the dry mass of a leaf unit of the $i$th woody species and $a$ is a constant. Separate regressions were created for adult bulls and members of breeding herds, and the difference between the mass of leaves harvested by each group was tested across a range of leaf sizes using analysis of covariance (ANCOVA), with leaf size specified as the covariate in the model. The analysis was conducted using R version 4.1.2 (*R Core Team, 2021*), and the 'tidyverse' (*Wickham et al., 2019*), 'ggpubr' (version 0.6.0; *Kassambara, 2023a*), and 'rstatix' (version 0.7.2; *Kassambara, 2023b*) packages. A leaf unit was defined as the smallest unit held by the trunk and was most often represented by a leaflet. The data used to create the regression models were collected by observing adult bulls ($n = 17$) and members of breeding herds ($n = 18$) stripping or plucking leaves from a range of woody plant species. When an elephant moved off, five samples were gathered from the plant it had been feeding on by simulating the amount of leaf it had extracted at each trunkload. The average dry weight of the five samples provided an estimate of $S_{leaf}$. In this way, estimates of $S_{leaf}$ were obtained for a range of plant species, across a number of different adult bulls and members of breeding herds. The average dry mass of a leaf unit was calculated for each woody plant species by randomly selecting five individual plants of each species and collecting between 10 and 100 leaf units, depending on the size of a unit, from each individual, and drying them to constant mass. The average mass of a trunkload of green leaves for adult bulls was then estimated for each landscape unit as:

$$S_{leaf,bull} = \sum_{i=1}^{n} S_{leaf,i,bull} D_{i,bull},$$

where $S_{leaf,bull}$ is the dry mass of a trunkload of green leaf gathered by an adult bull from the $i$th woody species and $D_{i,bull}$ is the density (no. individuals ha$^{-1}$) of the $i$th woody species with canopy volume below six m and >25% canopy volume with green leaf. $S_{leaf,cow}$ was calculated in the same way except $S_{leaf,bull}$ was substituted by the value for members of breeding herds and $D_{i,bull}$ was replaced by $D_{i,cow}$ which was the density (no. individuals/ha) of the $i$th woody species with canopy volume below four m and >25% canopy volume with green leaf (see *Clegg & O'Connor, 2017* for how density was calculated). Heights of six m and four m were used as cut offs because these represent the maximum feeding heights for adult bulls and adult cows respectively (*Clegg, 2010*). Only woody plants with >25% green leaf were considered available for foraging because elephants avoid eating senescent leaves (*Clegg, 2010*).

### Estimation of trunkload mass for bark from canopy branches
Elephants remove bark from the canopy branches of shrubs and trees by breaking off a branch, placing it in the mouth and then chewing off the bark along the length of the

branch. The average mass of bark, $W$, extracted from a branch was estimated for the most commonly used species in the following way. For each species, 10 to 13 branches, across a range of diameters and lengths, were collected. Each branch was gathered from a separate, randomly chosen plant. For each branch, the bark was removed, dried to constant mass, and weighed to the nearest gram. Once the bark had been removed, the length (cm) and diameter (mm) at 10 cm intervals was measured for each branch, from which an average diameter was calculated for each branch. Equations predicting the mass of bark from a branch were developed using Systat 9 (*SPSS, 1998*), for each species, by estimating the parameters of the following model using non-linear regression:

$$W = aLD + b(LD)^2,$$

where $W$ is the dry mass of bark, $L$ is the length of the branch, $D$ is the average diameter of the branch and $a$ and $b$ are constants. One hundred branches that had been fed on by adult bulls and 100 branches that had been fed on by members of breeding herds were collected for each plant species. For each branch, the length and average diameter was measured for the part from which bark had been removed. When a forked branch was debarked, each fork was measured separately. The mass of bark removed from each branch was then estimated using the regression equations. The average mass of bark removed from branches of each species was calculated for adult bulls and members of breeding herds by calculating the average of the respective samples. Differences between means were tested using an aligned ranks transformation ANOVA with *post hoc* multiple comparisons with Tukey's adjustment (chosen because data were non-normal and heteroscedastic). The analysis was conducted using R version 4.1.2 (*R Core Team, 2021*) and the 'ARTool' package (version 0.11.1; *Kay et al., 2021*).

The average trunkload mass of bark for bulls, $S_{bark,bull}$, was then calculated for each vegetation type as follows:

$$S_{bark,bull} = \sum_{i=1}^{n} W_{i,bull} D_{i,bull},$$

where $W_{i,bull}$ is the average dry mass of bark removed by bulls for the $i$th species, and $D_{i,bull}$ is the relative density of the $i$th species with canopy below six m. $S_{bark,cow}$ was calculated in a similar way except $W_{i,bull}$ was replaced by $W_{i,cow}$ and $D_{i,bull}$ by $D_{i,cow}$ which was the relative density of individuals of the $i$th species with canopy below four m.

## Running the models

The models for each forage type were run in Idrisi GIS software (*Eastman, 2001*) at a daily time step for one annual cycle (12th of March 2002 to the 10th of March 2003). Predictor variables, which were constructed by adding attribute data to a raster map of the 65 land units, were included in the regression equations as raster layers. Once the models had been run, the mean, minimum and maximum mass potentially harvested per trunkload over the study area were calculated from the daily output raster layers, and timeseries plotted for each forage type. Daily differences in the mean mass harvested across all paired combinations of forage types were tested for significance using a Wilcoxon signed rank

test (assumptions for paired $t$-tests were not met). The analyses were conducted using R version 4.1.2 (*R Core Team, 2021*). Comparisons were conducted separately for forage harvested by adult bulls and members of breeding herds, and for the differences between adult bulls and members of breeding herds.

## RESULTS

### Model parameterization
#### *Estimation of trunkload mass for grass and forbs*

The height of green grass, mixed green and dry grass, and forbs varied seasonally and across landscape units (Fig. 1). The height of grasses and forbs decreased during the dry season due to herbivory, trampling and senescence. Within the same land unit, mixed grass was usually taller than green grass, and grasses were generally taller than forbs.

When feeding on *Setaria incrassata*, a robust, perennial grass, the mean tuft circumference utilised by adult bulls and members of breeding herds was 35.5 cm and 22.3 cm, respectively, compared to 52.7 cm for *Setaria* tufts in the plant community (Fig. 2). A Kruskal–Wallis test, followed by a Dunn *post-hoc* test showed significant differences between the three groups ($\chi^2 = 122.34$; $df = 2$; $P = 2.2\text{e}{-}16$; adult bull *vs* member breeding herd, $Z = -8.898$, $P = 1.14\text{e}{-}18$; adult bull *vs* plant community, $Z = -2.398$, $P = 1.65\text{e}{-}02$; member breeding herd *vs* plant community, $Z = -9.614$, $P = 2.1\text{e}{-}21$), and consequently the mean values for adult bulls and members of breeding herds were used to set upper size limits for tuft utilisation.

In each landscape unit, there was a strong linear relationship between the mass of a grass plant and the product of leaf height and tuft basal area (average $R^2_{\text{adj}}$ ($\pm$ SD) $= 0.74 \pm 0.12$) (Fig. 3).

Sixty-eight species of forb were included in the analysis with the average dry mass of a fully developed plant ranging from two g to 418 g. The weighted average dry mass of a fully developed forb plant (assumed to represent the maximum mass of a trunkload) growing in a landscape unit ranged from 14 g to 160 g, with a mean of 62.7 g (Fig. 4). Sixty per cent of the landscape units had an average forb dry mass between 40 and 80 g.

#### *Estimation of trunkload mass for leaves from woody plants*

There was a positive linear relationship between the mass of a trunkload of leaves harvested from woody plants and the mass of an individual leaf unit (Fig. 5). Adult bulls harvested heavier trunkloads of leaves than members of breeding herds (ANCOVA, with dry mass of a leaf unit as the covariate: $F_{(1,32)} = 8.262$; $P = 0.007$).

#### *Estimation of trunkload mass for bark from canopy branches*

The mass of bark removed from canopy branches was successfully modelled for *Colophospermum mopane*, *Grewia bicolor* and *Grewia monticola* (the most commonly utilised species for canopy bark) using quadratic regression equations with the product of the length and diameter of the debarked branch as the predictor variable (Fig. 6). The aligned ranks transformation ANOVA to test for differences in the mass of bark harvested by adult bulls and members of breeding herds across the three plant species showed a

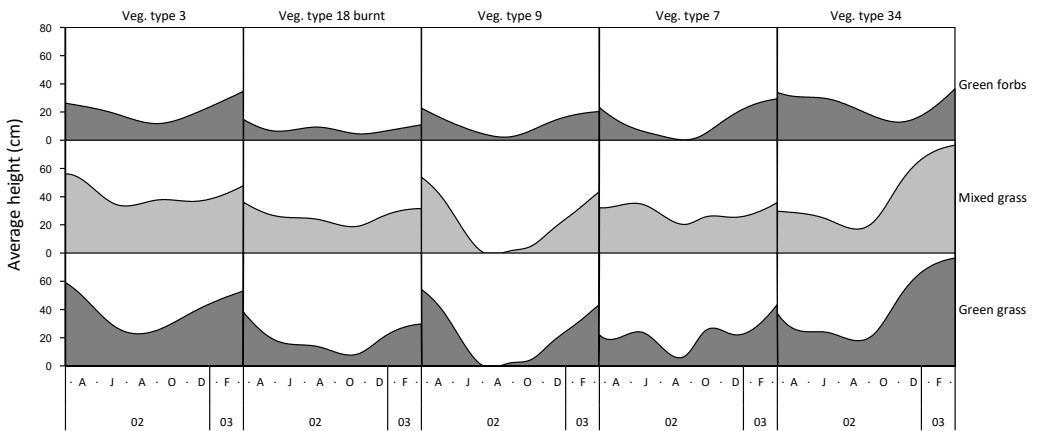

**Figure 1** **Examples of variation in the height of green grass, mixed grass, and green forbs across five landscape units between March 2002 and March 2003.** Vegetation type 3 = *Julbernardia globiflora–Strychnos madagascariensis* tall woodland; Vegetation type 18 = *Colophospermum mopane–Courbonia glauca* shrub open tall woodland; Vegetation type 9 = *Acacia tortilis* open woodland; Vegetation type 7 = *Albizia petersiana–Strychnos potatorum* shrub open woodland; Vegetation type 34 = *Dichrostachys cinerea–Dalbergia melanoxylon* open woodland.

significant effect for elephant group (F(1,630) = 81.69; P < 2.22e−16), plant species (F(2,630) = 3.84; P = 0.022), and for the interaction between the elephant group and plant species (F(2,630) = 6.56; P = 0.0015). Adult bulls harvested heavier trunkloads (25.6–41.1 g) than members of breeding herds (19.8–22.1 g) for all comparisons between the plant species (P < 0.001), but there were no significant differences between the mass harvested across the plant species for adult bulls or members of breeding herds (Fig. 7).

## Model outputs

The average mass potentially harvested per trunkload changed seasonally for green grass, mixed grass, green forbs, and green leaves but was assumed constant over the annual cycle for canopy bark (Fig. 8). The mass of a trunkload of green grass and green forbs was three times heavier during the late-wet season than at the height of the dry season for both adult bulls and members of breeding herds (Table 1). Trunkload mass was also heavier during the wet season for mixed grass and green leaves from woody plants but only by a factor of one and a half. The variation in trunkload mass across the study landscape was greater for the herbaceous forage types than those from woody plants. For example, for adult bulls during the late wet season there was a difference of 218 g between the minimum and maximum estimates of a trunkload of green grass but only a 33 g difference for a trunkload of green leaves from woody plants. The variation between minimum and maximum estimates was greater for adult bulls than for members of breeding herds.

Grass and forbs yielded heavier trunkloads than leaves and bark from woody plants during the wet season for both adult bulls and members of breeding herds (Wilcoxon signed rank test, P < 0.01), but this was reversed in the dry season (Figs. 9 and 10). Of the herbaceous forage types, green grass produced the heaviest trunkloads during the late-wet

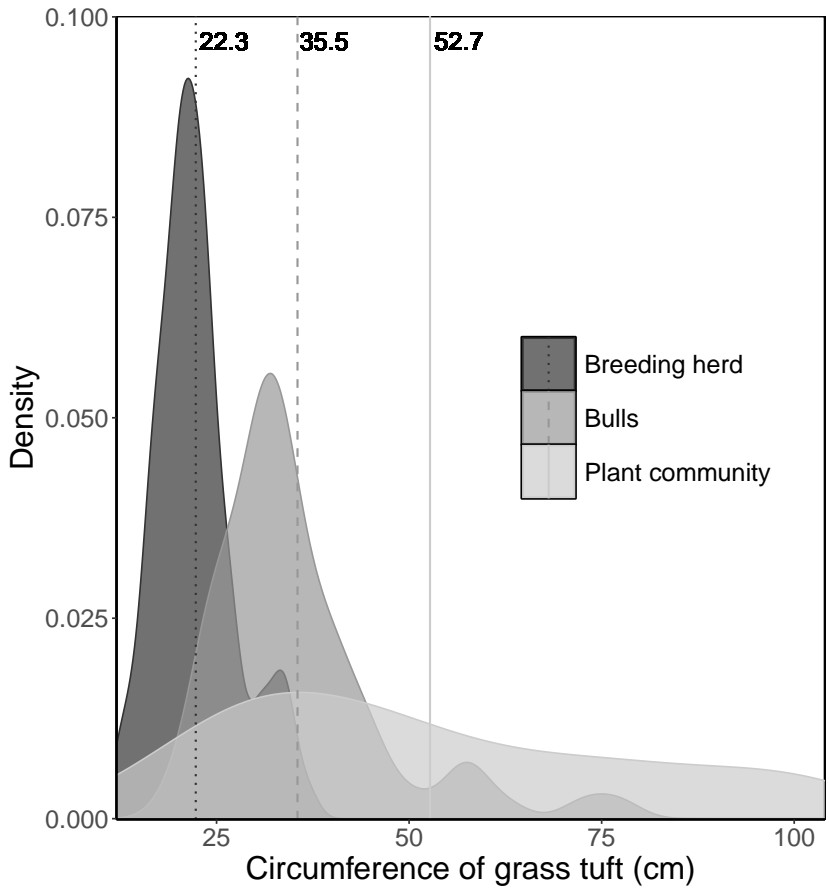

**Figure 2** Kernel density plots comparing the size distribution of *Setaria incrassata* tufts growing in a *Setaria incrassata* grassland that were utilised by adult bulls and members of breeding herds. Vertical lines denote the mean tuft circumference for each category. Differences between means were significant (Kruskal–Wallis test: $\chi^2 = 122.34$; $df = 2$; $P = 2.2e-16$. Dunn *post hoc* test: adult bull *vs* member breeding herd, $Z = -8.898$, $P = 1.14e-18$; adult bull *vs* plant community, $Z = -2.398$, $P = 1.65e-02$; member breeding herd *vs* plant community, $Z = -9.614$, $P = 2.1e-21$).

season but trunkloads of mixed grass and green forbs were similar or heavier than green grass in the early- and mid-wet season.

Differences between adult bulls and members of breeding herds were apparent. Firstly, trunkloads were heavier for adult bulls than members of breeding herds across all forage types (Wilcoxon signed rank test, $P < 0.01$), except for forbs, which were assumed to yield a similar mass for both groups (Fig. 11). Secondly, trunkloads of grass were noticeably heavier than trunkloads of forbs for adult bulls, while grass and forbs yielded a similar mass for breeding herds. Thirdly, trunkloads of canopy bark were significantly heavier than trunkloads of forbs for a large portion of the dry season for adult bulls but not for members of breeding herds. Lastly, during the wet and cool-dry seasons, green leaves yielded heavier trunkloads than canopy bark for members of breeding herds but the two forage types yielded a similar mass for adult bulls over the same periods.

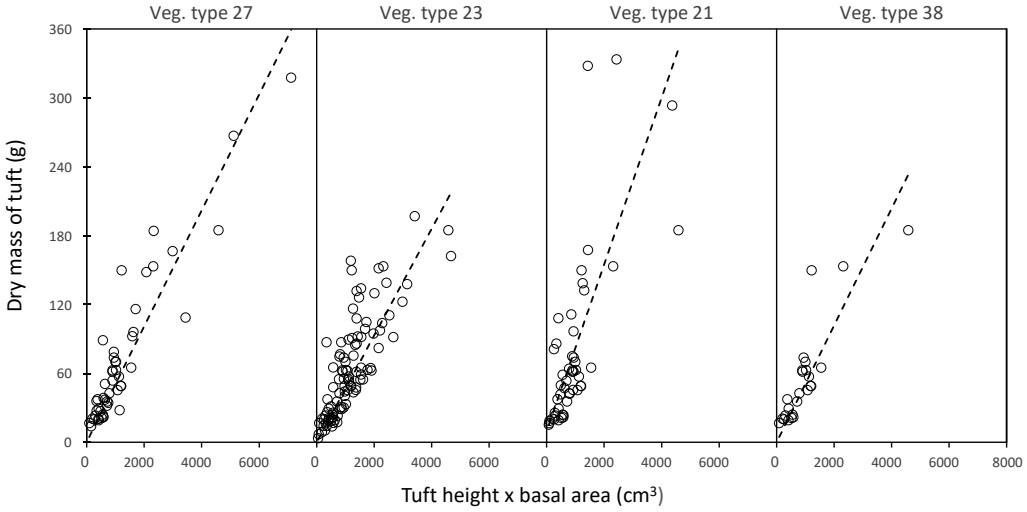

**Figure 3 Examples of the linear relationship between the mass of a grass tuft and the product of height and basal area in five selected landscape units.** Vegetation type 27 = *Acacia nigrescens–Combretum apiculatum* shrub open woodland, mass = 0.050541 (height × basal area), $R^2_{adj} = 0.83$; Vegetation type 23 = *Colophospermum mopane–Enneapogon scoparius* open woodland, mass = 0.050016 (height × basal area), $R^2_{adj} = 0.62$; Vegetation type 21 = *Colophospermum mopane–Terminalia prunioides* dense shrub open tall woodland, mass = 0.072608 (height × basal area), $R^2_{adj} = 0.50$; Vegetation type 38 = *Colophospermum mopane–Endostemon tenuiflorus* open woodland, mass = 0.050897 (height × basal area), $R^2_{adj} = 0.70$.

## DISCUSSION

The mass of forage harvested with each bite is an important determinant of the rate of food intake by mammalian herbivores (*Shipley, 2007*). Bite mass increases with body size (*Shipley et al., 1994*), and because African elephants are the largest land mammals, their bites are expected to be heavier than those of other browsers and grazers. Elephants harvest food with their trunks, so for the purpose of this discussion we use trunkload mass instead of bite mass when referring to elephants.

The average maximum mass of a trunkload of grass that elephants could potentially harvest across the study landscape was 93 g for adult bulls and 58 g for members of breeding herds (Table 1), which is more than 75 times heavier than the bite mass reported for other grazing herbivores (*e.g.*, cattle = 0.77 g, white rhinoceros = 0.75 g, bison = 0.6 g, sheep = 0.18 g (*Gordon, Illius & Milne, 1996*; *Rook et al., 2004*; *Shrader, Owen-Smith & Ogutu, 2006*; *Raynor, Joern & Briggs, 2015*; *Boval & Sauvant, 2021*)). In comparison, our predictions for trunkloads of browse (adult bulls = 37 g, members of breeding herds = 29 g) were only 8 times heavier than the bite mass reported for other browsers (*e.g.*, male giraffe = 3.3 g, moose = 3.2 g, female giraffe = 2.2 g, kudu = 0.45 g, goat = 0.18 g, impala = 0.14 g (*Pellew, 1984*; *Cooper & Owen-Smith, 1986*; *Haschick & Kerley, 1997*; *Pastor et al., 1999*)). We recognise that some authors classify forbs as browse but here we refer to browse as leaves from woody plants. We could not find any estimates in the literature for the dry mass of trunkloads of grass, but the estimate of 35 g for browsing elephants (*Schmitt, Ward & Shrader, 2016*) falls within our range of 16–37 g, giving support to our model for browse.
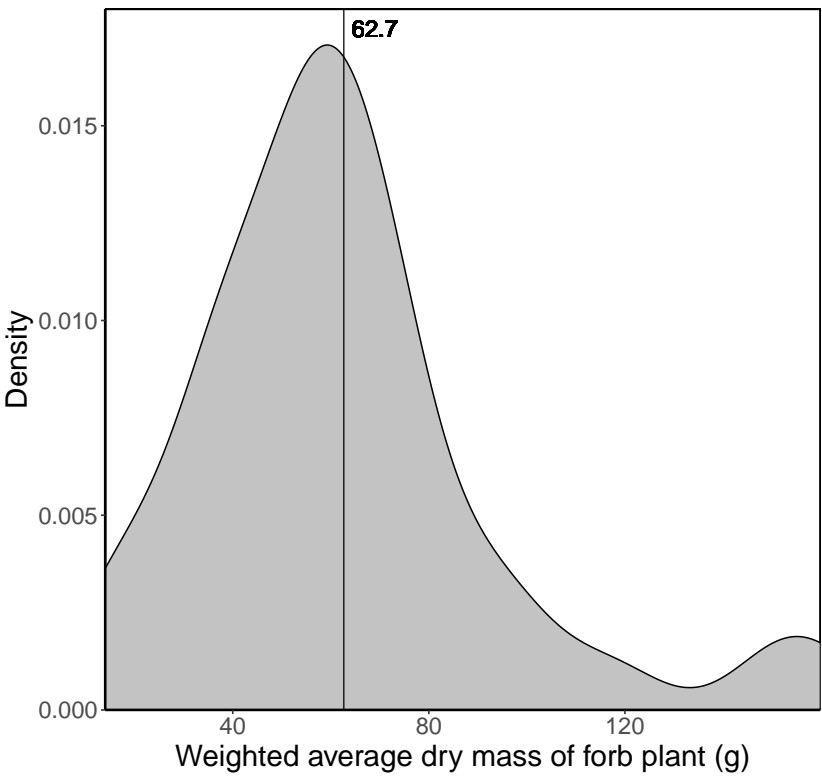

**Figure 4** **Kernel density plot showing the distribution of the average weighted dry mass (g) of a fully developed forb plant across the 65 landscape units.** The vertical line denotes the average mass (62.7 g) across the landscape units.

Our models suggest that elephants extract a heavier mass of food than other herbivores with each harvesting action, but the increase was larger than expected from body size. *Shipley et al. (1994)* used the maximum bite mass of 34 herbivore species, excluding elephants, to develop a relationship between bite mass and body size. Separate regressions were created for grazers and browsers. When feeding on grass, their equations predict a bite mass of 4.4 g for adult elephant bulls and 2.9 g for cows, and 13.7 g and 8.8 g, respectively when feeding on browse (assuming a body mass of 4,850 kg for bulls, and 3,500 kg for cows (*Pretorius et al., 2016*)). This is 20-times lower than our predictions for grazing adult bulls and members of breeding herds, and three times lower for both groups when feeding on browse.

Body size influences bite mass because bite volume, which is the product of the depth and gape area of the mouth, increases with body size (*Pretorius et al., 2016*). However, field measurements of bite volume for elephants are considerably larger than that predicted by body mass (*Pretorius et al., 2016*). This is because the mass of forage harvested by elephants is not determined by mouth size but rather by the ability of the trunk to grasp and bundle forage into the buccal cavity (*Owen-Smith, 1988*; *Pretorius et al., 2016*). Our results suggest that the trunk is most effective when elephants are feeding on grass. This is because the trunk can be easily wrapped around the long, linearly shaped leaves, and an entire tuft

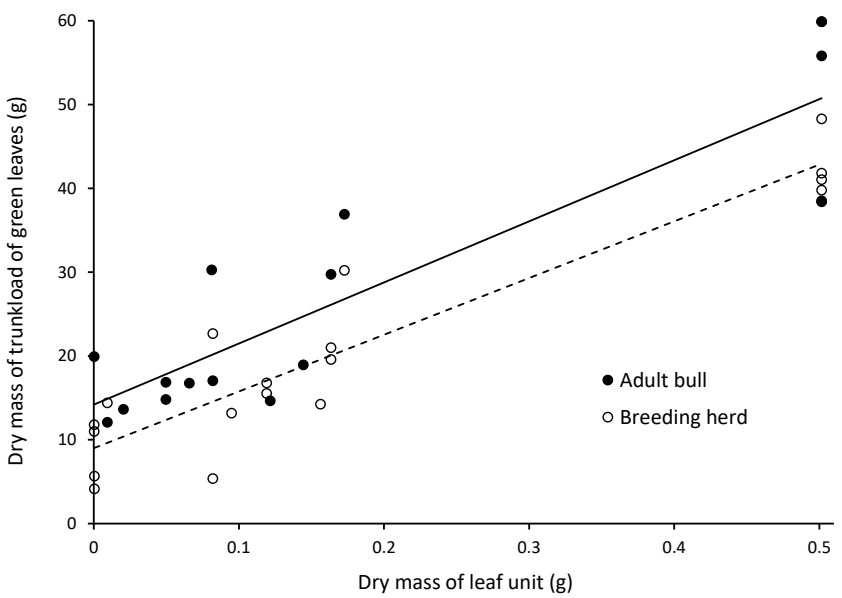

**Figure 5** **Linear relationships between the dry mass of a trunkload of green leaves and the mass of a leaf unit for adult bulls and members of breeding herds.** Adult bulls (solid line, $Y = 72.902X + 14.196$, $R^2_{adj} = 0.79$) harvested heavier trunkloads of leaves than members of breeding herds (dashed line, $Y = 67.669X + 9.001$, $R^2_{adj} = 0.87$). ANCOVA, with the dry mass of a leaf unit as the covariate: $F(1,32) = 8.262$; $P = 0.007$.

uprooted or the above ground parts harvested, even if the grass is tall (*i.e.,* the volume of grass harvested is not constrained by mouth volume). The harvested material is then bundled by the trunk and pushed into the mouth, which has the effect of compressing an otherwise large volume of food into a compact bite. In essence, the trunk considerably increases the mass of grass that can be harvested compared to other grazers that crop forage using their lips or tongues. This explains why elephants can harvest loads of grass that are up to 20-times heavier than the bite mass predicted from body size. The trunk provides a similar advantage when feeding on forbs because an entire plant can be plucked from the soil, regardless of plant height. However, during the wet season, the mass of a trunkload of green forbs harvested by adult bulls is often less than a trunkload of grass because forb plants seldom weigh more than fully developed grass tufts. The advantage provided by the trunk is less when elephants are browsing because leaves from shrubs and trees are more difficult for the trunk to handle. This causes many leaves to fall from the trunk's grasp during the harvesting process (*Clegg & O'Connor, 2016*), which lowers the mass of a trunkload when feeding on browse. However, the trunk still affords a three-fold advantage because a larger volume of leaves can be stripped off a branch than could otherwise be achieved using the mouth. Bark from canopy branches of shrubs and trees had the lowest harvestable mass because elephants extract this forage type by chewing the bark off the branches. The chewing of bark off the branches is therefore constrained by mouth volume and does not benefit from the trunk, although the trunk improves harvesting efficiency by reducing prehension time of these branches (*Clegg & O'Connor, 2016*). Unlike grasses,

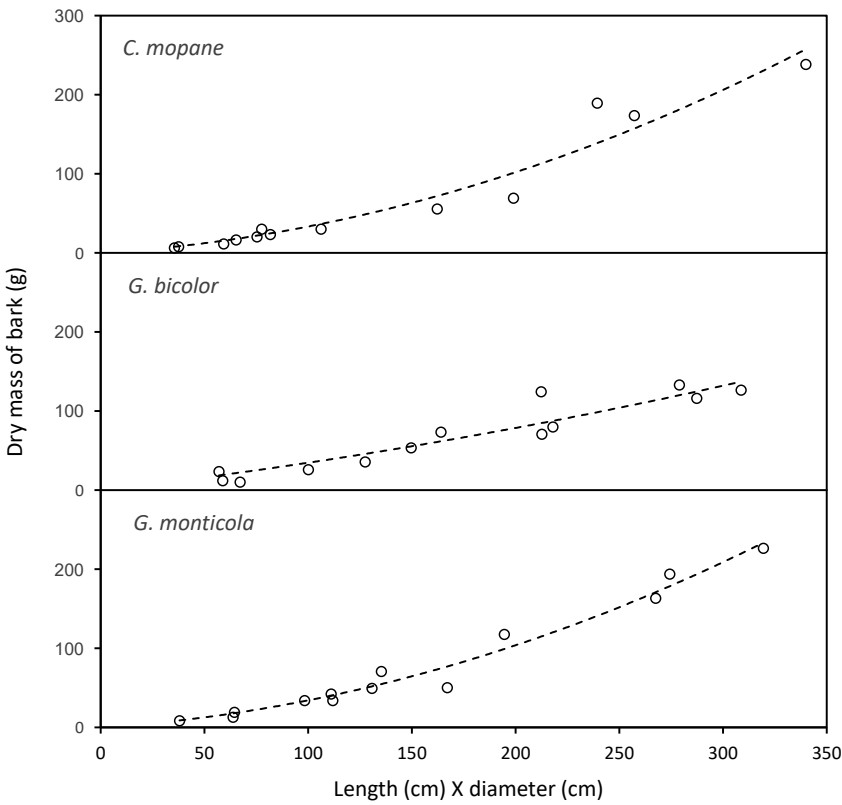

**Figure 6  The relationship between the mass of bark chewed off branches and the product of the length and diameter of the branches.** *Colophospermum mopane*: $Y = 0.164715X + 0.001772X^2$, $R^2_{adj} = 0.94$; *Grewia bicolor*: $Y = 0.299728X + 0.000467X^2$, $R^2_{adj} = 0.89$; *Grewia monticola*: $Y = 0.156027X + 0.001769X^2$, $R^2_{adj} = 0.94$.

woody plants and forbs are often chemically defended, and this might increase the degree of selectivity required and, in turn, reduce intake rate when elephants are feeding from these plant groups. However, it should be noted that our models suggest that even if woody plants and forbs were undefended chemically, elephants would harvest lighter trunkloads from them compared to what is achievable from grass. This because trunkload mass is constrained by the mass of a leaflet for woody plants and by the mass of the whole plant for forbs. In other words, the primary constraints to harvestable mass are independent of the level of chemical defence.

A larger trunkload mass (Table 1, Figs. 9 and 10) and therefore a potentially higher rate of intake when feeding on grass explains why elephants preferentially graze during the wet season (*Buss, 1961*; *Guy, 1976*; *Cerling et al., 2009*; *Codron et al., 2011*; *Shannon, Mackey & Slotow, 2013*; *Gill et al., 2023*), despite the abundance of browse at that time of year. In savanna environments, the advantage provided by grass is confined to the wet season because the height of green grass, which is an important determinant of trunkload mass, declines during the dry season due to herbivory, trampling, and senescence. This means that the harvestable mass of green grass declines, and a point is reached when it is

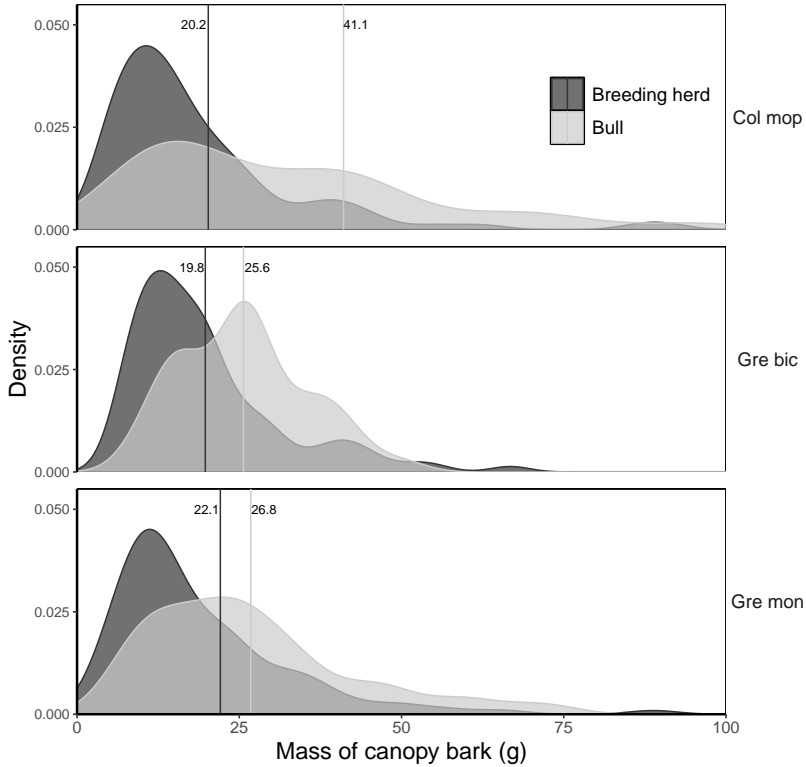

**Figure 7** **Kernel density plots comparing the distribution of the dry mass (g) of canopy bark harvested from *Colosphospermum mopane*, *Grewia bicolor*, and *Grewia monticola* by adult bulls and members of breeding herds.** Vertical lines denote the mean bark mass for each category. Differences between means were significant (aligned ranks transformation ANOVA, multiple comparisons with Tukey's adjustment: *P* < 0.001) for all comparisons between adult bulls and members of breeding herds, but not for comparisons between plant species within the adult bull and breeding herd categories.

superseded by the mass of green browse that can be harvested from shrubs and trees (Figs. 9 and 10). This explains the well-established trend of savanna elephants switching their diet from grass in the wet season to browse during the dry season (*Guy, 1976*; *Shannon, Mackey & Slotow, 2013*; *Gill et al., 2023*). However, the change in diet may not only be due to a decline in the harvestable mass of grass, but also to the reduced availability of green grass (*Clegg & O'Connor, 2017*). Elephants are potentially reliant on easily digestible cell solubles (*O'Connor, Goodman & Clegg, 2007*), which are not readily available in senescent forage. Furthermore, when grass is composed of a mixture of green and senescent material, it takes elephants twice as long to harvest a trunkload because the senescent component of the forage is first removed by thrashing the trunkload against a leg or the chest before the remaining material can be ingested (*Clegg & O'Connor, 2016*). The time to conduct this additional harvesting action reduces the intake rate and makes mixed dry and green grass a less profitable source of food (*Clegg, 2010*; *Clegg & O'Connor, 2016*). As the dry season progresses, the soil continues to dry, and the leaves of shrubs and trees begin to senesce. Drought-tolerant woody plants generally have smaller leaves that persist for longer into the

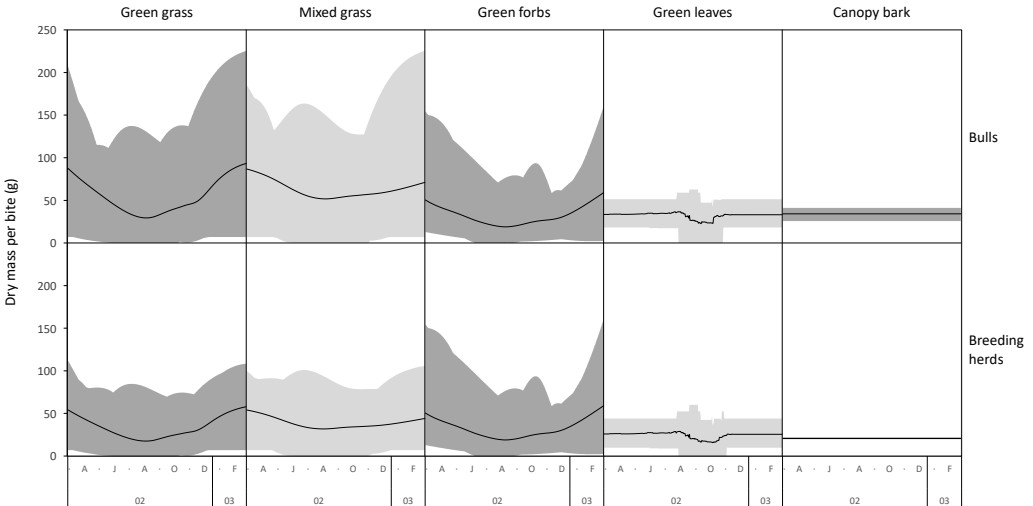

**Figure 8** Average (black lines), and maximum and minimum (grey shading) mass of a trunkload that adult bulls and members of breeding herds could potentially harvest for each forage type within the study landscape over an annual cycle (March 2002 to February 2003).

**Table 1** Maximum and minimum average mass (g) of a trunkload potentially harvested by adult bulls and members of breeding herds for each forage type over the study area.

| | Average mas per trunkload (g) | | | |
| --- | --- | --- | --- | --- |
| | **Maximum** | | **Minimum** | |
| **Forage type** | **Adult bull** | **Breeding herd** | **Adult bull** | **Breeding herd** |
| Green grass | 93.3 | 57.9 | 29.4 | 17.6 |
| Mixed grass | 86.8 | 54.0 | 51.7 | 31.9 |
| Green forbs | 59.0 | 59.0 | 18.9 | 18.9 |
| Green leaves | 36.8 | 29.0 | 22.8 | 15.8 |
| Canopy bark | 34.2 | 20.6 | 34.2 | 20.6 |

dry season than the larger leaves of broad-leaved species (*Midgley et al., 2004*). This causes the trunkload mass of green browse to decline as the dry season progresses. Eventually, a point is reached when the mass of a trunkload of bark exceeds the mass from browse (Figs. 9 and 10), which is consistent with reports that elephants eat larger amounts of bark in the late dry season (*Guy, 1976*; *Owen-Smith & Chafota, 2012*). The predictable seasonal change in the profitability rankings of the forage types is due to the different profile shapes of their harvestable mass timeseries. Grass and forbs have u-shaped profiles, while the profiles of leaves and bark are relatively flat. Consequently, from the end of the wet season the trunkload mass of herbaceous forage declines to a level below that of the more constant harvestable mass of browse and bark, only to increase again during the next wet season.

Seasonal changes in trunkload mass across forage types correlated well with reported seasonal changes in the diet of elephants, but it should be recognised that trunkload mass is only one of several constraints to intake rate, and that more holistic hypotheses for the

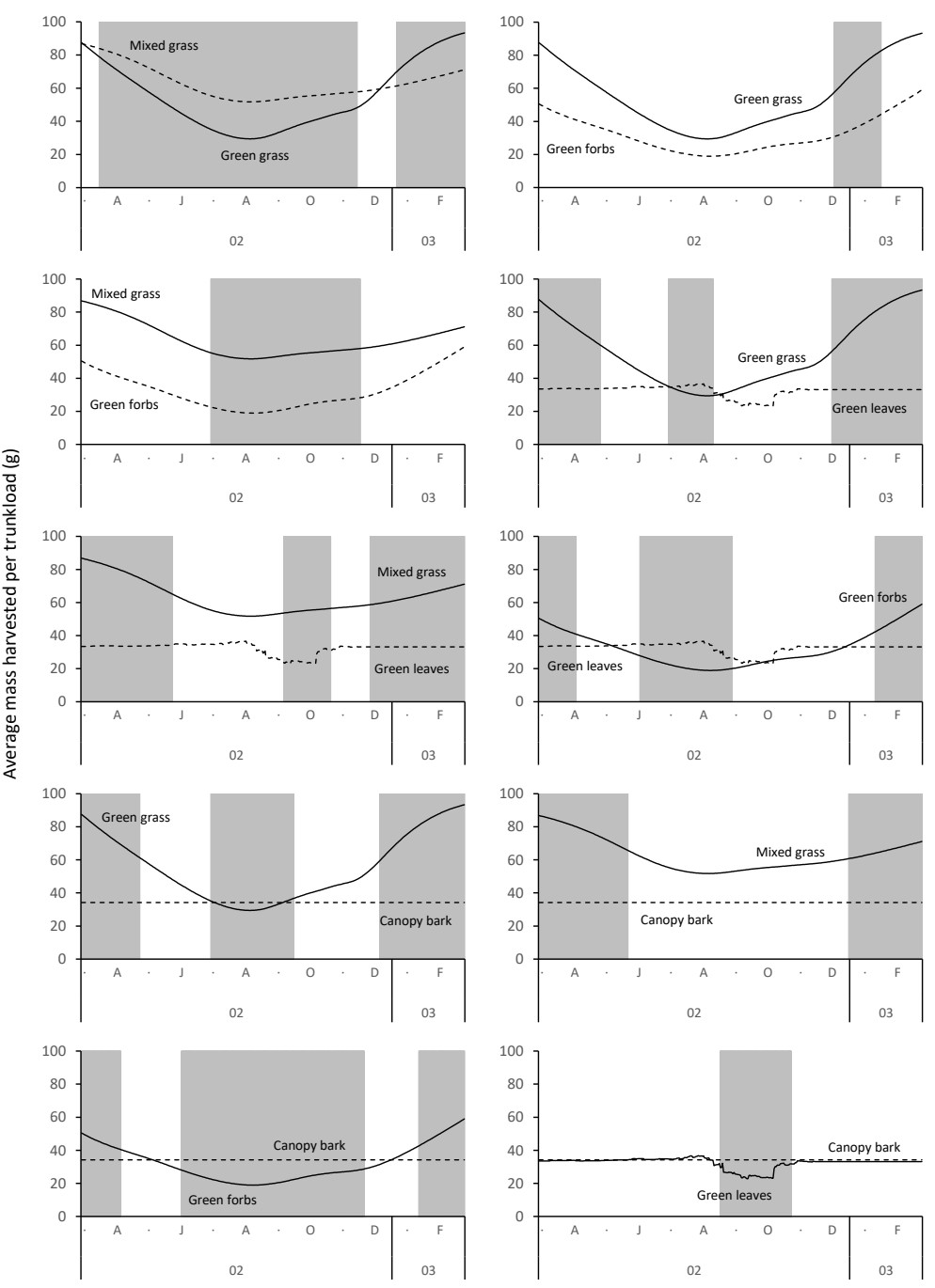

**Figure 9  Pairwise comparisons across forage types of the average mass of a trunkload that adult bulls could potentially harvest within the study landscape over an annual cycle (March 2002 to February 2003).** Periods of significant difference (Wilcoxon signed rank test, $P < 0.01$) between the forage types are shaded grey.

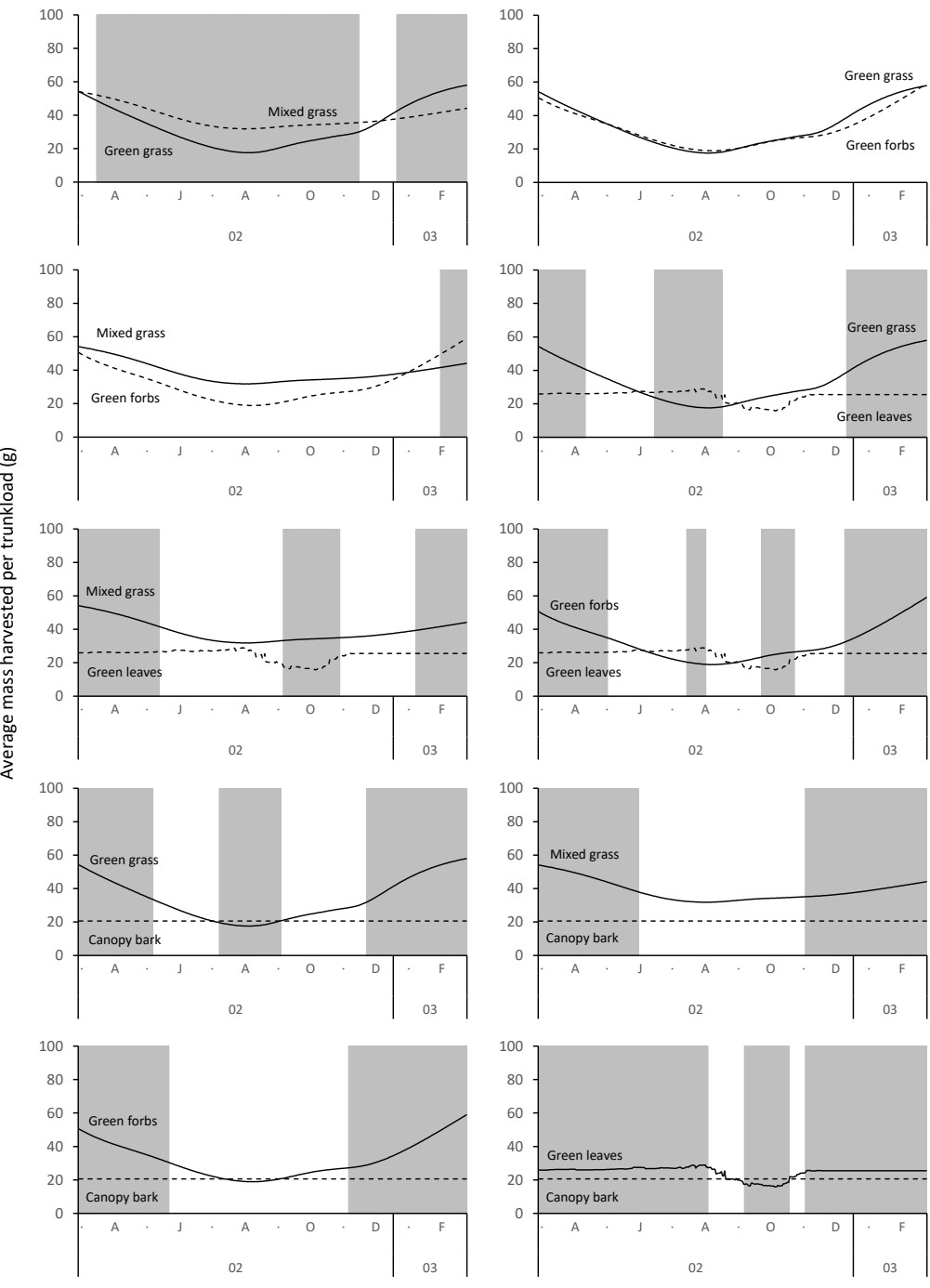

**Figure 10 Pairwise comparisons for selected forage types of the average mass of a trunkload that members of breeding herds could potentially harvest within the study landscape over an annual cycle (March 2002 to February 2003).** Periods of significant difference (Wilcoxon signed rank test, $P < 0.01$) between the forage types are shaded grey.

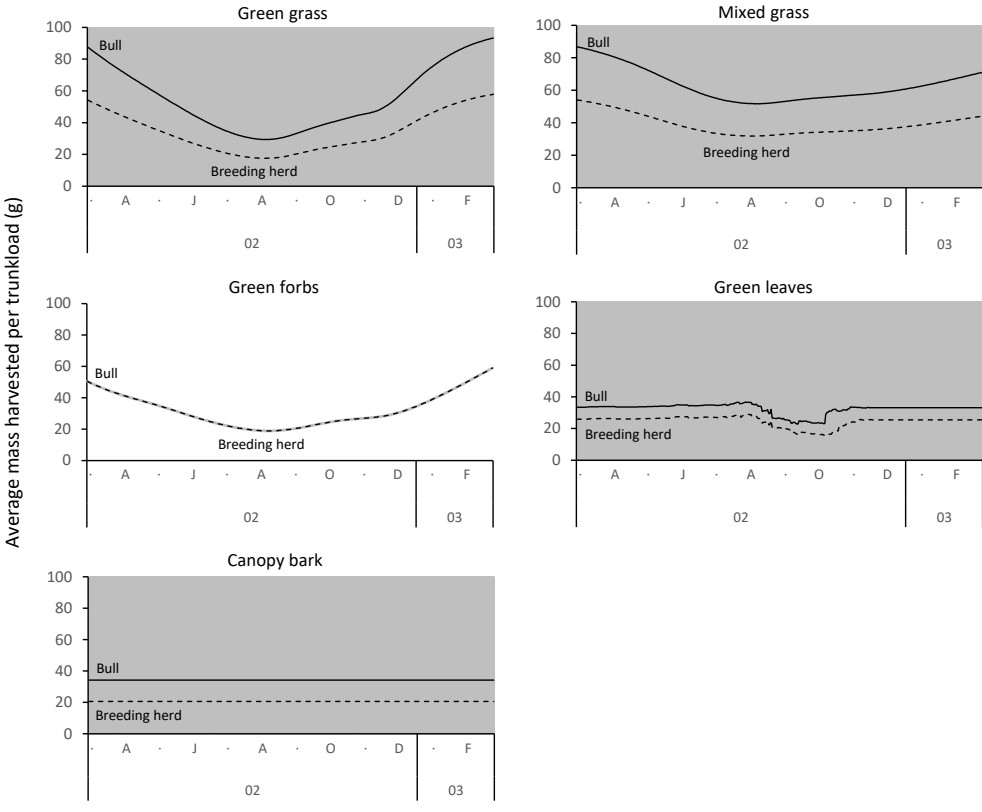

**Figure 11** **Pairwise comparisons across forage types of the average mass of a trunkload that adult bulls and members of breeding herds could potentially harvest within the study landscape over an annual cycle (March 2002 to February 2003).** Periods of significant difference (Wilcoxon signed-rank test, $P < 0.01$) between the elephant groups are shaded grey.

changes in diet would be derived from an integrated intake rate model that simultaneously incorporates the effects of the important constraints to intake such as searching, harvesting and chewing times, and the density of accessible nutrients in the forage (see *Clegg, 2010* for an integrated model). An integrated model may predict larger differences between forage types than a model based on trunkload mass alone because the time to harvest and chew a trunkload also varies significantly across food types (*Clegg & O'Connor, 2016*). That being said, the outputs of an integrated model are not dissimilar to the outputs of the models in this study. This suggests that trunkload mass is potentially the most important constraint to intake for elephants, with the other constraints only having a minor influence.

Differences between adult bulls and members of breeding herds were apparent. Adult bulls harvested heavier trunkloads than members of breeding herds across all forage types, except forbs; and bulls harvested disproportionately larger trunkloads of grass and bark compared to members of breeding herds (Table 1, Fig. 11). These differences are possibly explained by the greater chewing capacity of bulls. Adult bulls are double the body size of adult cows and therefore have larger, stronger jaws that enable them to efficiently chew large trunkloads, especially of fibrous forage such as grass and bark. Adult cows and subadults

of both sexes do not have the chewing power and are forced to harvest smaller trunkloads, especially when feeding on fibrous forage types. This may explain why members of breeding herds are reported to switch from grass to browse earlier in the season than adult bulls, and why adult bulls eat more grass and bark than members of breeding herds (*Clegg, 2010*; *Shannon, Mackey & Slotow, 2013*).

It has been hypothesised that an elephant's trunk increases bite volume and therefore bite mass (*Pretorius et al., 2016*). Our results support this hypothesis, but we further suggest that the advantage of a trunk is greatest when feeding on grass, less when feeding on forbs and leaves from woody plants, and least when harvesting bark from branches of shrubs and trees. The trunk significantly increases bite mass above that predicted by body size and also allows simultaneous harvesting and chewing (*Clegg & O'Connor, 2016*), both of which increase the rate of food intake (*Clegg, 2010*). We posit that without a trunk elephants might be unable to achieve the level of food intake required to meet the needs of their exceptionally large body size.

The advantage provided by the trunk appears greatest when elephants are feeding on grass, and the effect is more pronounced for adult bulls than members of breeding herds. This suggests that green grass should be the preferred food type of adult bulls. This finding has potential implications for the relationship between elephants and woodlands because if adult bulls have access to green grass during the dry season (*e.g.*, in wetlands, reedbeds) their disproportionate impact on trees (*Guy, 1976*; *Hiscocks, 1999*) should be reduced because they will preferentially graze, allowing trees a respite from damage. Several authors have observed elephants preferentially grazing on green grass during the dry season (*Buss, 1961*; *Wyatt & Eltringham, 1974*; *Western & Lindsay, 1984*; *De Boer et al., 2000*; *De Longh et al., 2004*), and others have reported the buffering effect that access to green grass has on damage to woodlands by elephants (*Tinley, 1977*; *De Longh et al., 2004*). However, many of Africa's elephant populations are confined to semi-arid landscapes (*Huang et al., 2024*) where availability of green grass is limited to a few months during the rainy season, and wetlands are largely absent. We suggest that adult bulls living within these confines are locked into a diet composed of leaves and bark from woody plants for most of the annual cycle, which results in a high level of impact on woodlands. Historically, bulls would have been able to migrate to wetlands during the dry season, but this is now largely prevented by human interference (*Huang et al., 2024*).

Our models suggest that forbs are a potentially important source of food for elephants, with the potential mass harvested often superseding that from leaves and bark from woody plants. This prediction is supported by field observation, with elephants of both sexes being reported to eat large quantities of forbs when they are available (*Clegg, 2010*; *Clegg & O'Connor, 2017*). Ground-creeping and climbing forbs are particularly favoured possibly because their tissue has a high ratio of cell contents to fibre on account of climbers being held upright by the shrubs and trees they scramble over as opposed to a strengthening of their own cell walls, and ground-creepers not requiring strengthening support (*Clegg, 2010*; *Clegg & O'Connor, 2016*). However, despite the importance of forbs as a source of forage, they are often overlooked in studies of savanna vegetation (*Siebert & Dreber, 2019*). It should also be noted that the consumption of forbs complicates the use of carbon isotope

analysis to determine the diet of elephants. Most forbs and woody plants utilise the C3 photosynthetic pathway (*Codron et al., 2005*) and therefore they produce the same isotope of carbon during photosynthesis. Consequently, the relative contribution of herbaceous forage (which includes many forbs) and woody vegetation in the diet cannot be strictly determined from the ratio of carbon isotopes in ivory, teeth, or tail hairs. However, the relative contribution of savanna grasses, which utilise the C4 photosynthetic pathway, can be reliably determined.

## CONCLUSION

The strong correlation between the outputs of our models and the well-established trends in the seasonal changes in the diet of elephants suggests that elephants are preferential foragers of the largest trunkload on offer. This makes them preferential grazers when suitable green grass is available, and browsers when grass forage is scarce. An integrated model that incorporates the other constraints to intake rate results in a level of fine tuning (*Clegg, 2010*), but this study has shown that the seasonal change in trunkload mass can, on its own, account for general trends in the diet of African savanna elephants. The mass of a trunkload appears to be at the centre of their foraging decisions, which is consistent with the hypothesis that elephants use the rate of food intake to rank the profitability of forage types. Green grass provides adult bulls with a disproportionately large trunkload mass and therefore the availability of suitable green grass is predicted to buffer woodlands from heavy impact by adult bulls. Unfortunately, many of Africa's elephant populations are confined to semi-arid environments where green grass is available for only a limited period during the rainy season. We suggest that has contributed to the destruction of woodlands in many of Africa's protected areas with high densities of elephants.

## ACKNOWLEDGEMENTS

The authors thank Julius Matsuve, Julius Shimbani, Cryson Chinondo, and Philmon Chivambu for assisting with data collection in the field.

### Funding
This work was funded by The Malilangwe Trust, Chiredzi, Zimbabwe. The funders had no role in study design, data collection and analysis, decision to publish, or preparation of the manuscript.

### Grant Disclosures
The following grant information was disclosed by the authors:
The Malilangwe Trust, Chiredzi, Zimbabwe.

### Competing Interests
The authors declare there are no competing interests.

## Author Contributions

- Bruce W. Clegg conceived and designed the experiments, performed the experiments, analyzed the data, prepared figures and/or tables, authored or reviewed drafts of the article, and approved the final draft.
- Timothy G. O'Connor conceived and designed the experiments, authored or reviewed drafts of the article, and approved the final draft.

## Data Availability

The data is available at Zenodo: Clegg, B. (2024). Mass harvested per trunkload as a constraint to forage consumption by the African elephant [Data set]. Zenodo. https://doi.org/10.5281/zenodo.13839059.

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
