# Peer review of "Mass harvested per trunkload as a constraint to forage consumption by the African savanna elephant (Loxodonta africana)"

_PeerJ, doi:10.7717/peerj.19033_

## Round 0.1 · original submission · Minor Revisions

With a range of reviewer recommendations from Accept to Minor Revisions to Major Revisions, as editor I feel Minor Revisions is appropriate. This is because Reviewer 2 (Major Revisions) is primarily asking for a change in emphasis in some wording, rather than contesting methodology or results. A revised MS should take Reviewer 2 into account, and address in detail the comments of Reviewer 3.

·

Basic reporting

The writing is clear and unambiguous. The literature utilised is correct and sufficient. Structure is correct. Results link to hypotheses.

Experimental design

Experimental design is excellent. research questions are clear. Very rigorous investigation. Methods described well.

Validity of the findings

Findings are solid, robust and sound. Conclusions well stated.

Additional comments

In this paper, the authors used mechanistic models to estimate how mass harvested per trunkload of different vegetation types (e.g. grass, forbs, woody vegetation) varied between seasons, and males and breeding herds. They found that harvestable mass varied seasonally, and bulls harvested larger trunkloads compared to breeding herds across all vegetation types. In addition, elephants obtained trunkload masses of green grass that were >75 times larger than other grazers, and trunkload masses of woody vegetation that were 8 times larger than other browsers. This is not too surprising (but still very cool) as elephants are unique in this comparison in being able to use their trunks to obtain food and not having to rely on mouth width, teeth, tongues, or lips to take bites. In addition, I really liked the fact that the authors could quantify when the transitions would and should take place by determining when the harvestable mass of a preferred food source (e.g. green grass) declines to the point where it overlaps with the mass available next preferred food source (e.g. green browse). This provides a really nice example of how herbivores expand their diet breadth. Moreover, the similar patterns of forage selection between the authors’ models and typical elephants foraging patterns highlights elephants’ preference for green grass. Understanding this, the authors suggest that in areas with a high availability of green grass into the dry season would reduce elephant foraging and thus impacts on woody vegetation. Overall, I really enjoyed the paper. It was a real pleasure to read. It is well written, easy to follow, and timely. Moreover, the results will be helpful to anyone working on elephant foraging. Due to the high quality of the manuscript I cannot think of any adjustments that would improve the manuscript. This is the first time that this has happened, so congratulations to the authors!

Prof Adrian M Shrader
University of Pretoria

Reviewer 2 ·

Basic reporting

No comment

Experimental design

The design was clear and the approach robust.

Validity of the findings

No comment

Additional comments

This manuscript provided a comprehensive study of African elephant foraging behavior in the context of harvest rate (i.e. trunkload mass). The overarching aim of this study was to understand how variation in trunkload mass might influence food intake rate. The authors modelled the seasonal changes in the mass harvested per trunkload for the main forage types used by elephants across a spatially heterogenous savanna landscape to understand how variation in trunkload mass might influence food intake rate, and to parameterize the trunkload mass component of an integrated intake rate model for elephants. The authors specifically asked three questions: (1) does the mass harvested per trunkload differ across forage types, (2) does it change seasonally, and (3) does it differ between adult bulls and members of breeding herds?

I appreciate the findings of this paper and its contribution to this body of knowledge. The methods were appropriate and results interesting. I do however, have some suggestions that I believe would strengthen the manuscript.

General comments:
1. I did not feel like the framing of this manuscript was as strong as it could have been. Upon initial reading, the framing of this paper feels centered around the fact that elephants are ecosystem engineers and reshape their ecosystems via their foraging behavior (e.g., 2/3 of the Background section of the Abstract and the opening few paragraphs of the Introduction (L30-60)). Thus, the expectation with this level of detail led me to think that this paper was going to explore how foraging decisions would contribute to ecosystem engineering of elephants. However, while progressing through the rest of the Manuscript, the level of detail and emphasis placed on the ecosystem engineering of elephants that was clearly outlined in the Abstract in Introduction, fades away. Instead, the majority of the Discussion is focused on elephant foraging mechanics and trunk functionality, dietary preferences across seasons and demographics, and the role of specific forage types. The Discussion around foraging mechanics and trunk functionality is extremely interesting, especially when paired with discussion about the variation in harvest rate across seasons, demographic groups, and common vegetation types. Thus, I suggest toning down the detail and reframing the Introduction of this MS to be more in alignment with the way the Discussion is outlined. With this restructuring, I believe this will significantly improve the impact of this manuscript.

2. In the Discussion, the authors make statements stating that the advantage of a trunk is greatest when feeding on grass, and less when feeding on forbs and leaves from woody plants. While this is true, one point to consider is that woody plants are chemically defended and grasses are typically not and this advantage might only possible because elephants do not have to be as selective with grasses as they have to be with forbs/woody plants due to the varying level of chemical defenses that are in these species, and thus can harvest large amounts of grasses rapidly. This might be an important point to discuss.

Specific comments:
1. L36: I suggest rephrasing to say “pressure from other herbivores”
2. In the Methods, I suggest being up-front about when this study occurred. The years/months should be clearly outlined early on, and not buried in a clause (i.e., L121).
3. L121: is the word “under” a typo?
4. L134-137: Later in the manuscript, I see that you limited vegetation availability estimates to <6m. However, I was wondering about this point when I read the statements around L134-137. It might be good to include that you accounted for availability of vegetation within the reach of an elephant’s trunk in this part of the text.

·

Basic reporting

No comment

Experimental design

no comment

Validity of the findings

My only concern was that the authors should provide some support for their assumption that elephants succeed in uprooting the entire tuft and roots in dry soil with different robust perennial grasses. Their models assume that elephants always uproot the tuft, which is an assumption that could lead to over estimates of trunkload if they only succeed in uprooting the tufts of robust perennials say 70% of the time. They need to provide some support for their assumption.

Additional comments

This is a very important study with some key insights that advance our understanding of why elephants switch their diet from grass/forbs to browse to bark/roots over the annual cycle.
Some key findings of the study is (1) showing that the bite mass (trunkload) on grass and browse by elephants is much larger than predicted by Shipley’s regressions and (2) showing that the much higher trunkload of grass than browse explains why elephants prefer feeding on grass during the wet season but that this advantage declines as grasses senesce during the dry season and height and quality declines and bite mass on browse becomes larger, with bark becoming the largest late in the dry season. These findings provide a new understanding of why elephants switch their diet over the annual cycle.

Other comments:
Line 147. Do elephants manage to regularly uproot tufts of robust perennials such as Themeda triandra, Eragrostis rigidior, Cenchrus ciliarus, Setaria incrassata, etc? I suppose when the soil is wet that is easy enough, but when the soil is dry how often do they just rip off the leaves and stems, leaving the bases and roots behind? I would think that in dry soil the leaves and stems might break off before the tuft is uprooted. Merely stating that “elephants generally uproot and consume a whole tuft” needs to be backed up with evidence. This is important to quantify because if elephants don’t always succeed in uprooting the tufts of robust perennial grasses then your estimate of trunkload is going to be exaggerated, which might at least partially contribute to your finding that bite mass is 75 times heavier than for other grazers.
Figure 9. I am surprised to see the amount of green grass still available during the dry season. How is that possible considering that one would not expect any green grass during the late dry season in dryland savanna’s without floodplains? In my observations is the only habitat in semi-arid savannas where green grass persists throughout the dry season is wetland habitats, such as floodplains and dambo’s. For example, in Figure 9 green grass declines until August and appears to start rising again in late August. That could only happen if you got rain in August and that should surely be a serious anomaly, distinct from most years. I realize that in your region there is a possibility of getting some small winter rains called “gutu” but how common is that and was that the reason for the amount of green grass in this dry season (2002)? An explanation of why green grass is still so abundant (30g per trunkload) in the dry season and how representative the 2002 dry season rainfall was relative to most dry seasons would be helpful.

Table 1. Simply dividing the year into two seasons is too simplistic. The dry season should be split into the early dry season (May to July) when green grass is much more available than in the late dry season (August and September or even October depending on when the rains start in your region – in northern Botswana the rains almost never start in October). If the rains in your region start only in the latter half of October, then I would include the first half of October in the late dry season as this will be the hottest and most stressful time for herbivores and when I would expect the least amount of green grass and green leaves in the diet. You are obscuring key insights of how green grass and green leaves may decline to lowest levels in the late dry season, and how bark could become relatively the most profitable by the late dry season. For example, in Table 1 green grass is still higher than green leaves in the dry season, which defies your explanation of why elephants switch to green leaves. I reiterate here that I find it hard to understand why there is so much green grass in your models during the dry season.
Forbs are eaten as high-quality resources by a wide variety of browsers and even grazers (see work by Owen-Smith and Cooper for example). In other words, forbs provide the highest quality component of the diet of many browser mixed feeder species. The switch in the diet of elephants from forbs and grass in the wet season to browse in the early dry season and finally to bark in the late dry season represents shifting diet from a high-quality resource (forbs), staple resource (grass), reserve resource (browse) to a buffer/key resource (bark) (Owen-Smith 2002). A buffer resource is conceptualized to be a low-quality resource eaten only when high quality and reserve resources are depleted and functions to minimize loss of body condition during the late dry season, especially during droughts. Similarly, a reserve resource is of lower quality than a high-quality or staple resource and persists longer into the dry season because of greater soil moisture availability than the high-quality resource (trees have deeper root systems that access deep-layer soil moisture) – it provides adequate quality forage. You should discuss your findings in relation to this framework. For example, does grass provide better nutrition than browse on a gram for gram basis or is better nutrition gained by a grass diet purely by being able to eat much more grass than browse because of the trunkload mechanism, which is also made possible by the rapid passage digestive system of elephant? i.e. this may not be the case for a ruminant.
Something to note in your discussion however is that while elephants eat more grass than browse during the wet season, they do eat browse during the rainy season. The answer to this cannot be explained solely by the trunkload hypothesis because certain forbs will not provide much trunkload and as you show, browse provides a lower trunkload than green grass during the wet season, therefore, one would not expect elephant to eat any browse during the wet season. The answer may lie in the observation that forbs and browse have much better phytochemical amount and diversity than grasses and thereby play a key role as medicinal resources (see Fred Provenza papers). Thus, elephant may be eating certain forbs and browse during the wet season, not for nutritional purposes but for medicinal purposes (prophylactic and therapeutic). This may explain why they eat some browse and a high diversity of forbs during the wet season.

Reviewer 4 ·

Basic reporting

1.1. Text

The manuscript is clearly written, it is well articulated and fairly complete in terms of references. I find the introduction a bit long in describing the various processes that influence intake rate while only bite mass is then truly investigated.

1.2. Figures and tables

Figure 6: I think the equation for Mopane is not showing the correct R². Colophospermum mopane: Y = 0.164715X + 0.001772X2, R2
adj = 0.094 (I feel it should be 0.94, from what I gather from the fitted line)

For Figure 9 and 10 there are many instances where it is difficult to understand why th values are significant for a given month when compared to the NS values for other months. It may be that the sample size is irregular, hence the power changes by months, but to me it really suggests that one cannot conclude firmly on such small variation.
A specific example from Fig 10: It is difficult to think that the difference between green leaves and green grass in July-August 02 is more significant than in May or October 02 (NS on the graph). The same applies for Green grass and canopy bark in August, displayed as significant (shaded grey) but the curves are son close compared to June, or November, though the graphs suggests these are not significant.

Y legend of Figure 10 is incomplete


1.3. I do not see where the raw data is supplied

Experimental design

The MS reports an impressive set of field measurements to be able to calibrate the trunk loads. The trunk loads in the study are an index of the achieved benefits associated with foraging in a specific habitat and on a specific resource for a given season. There is an ambiguity in the introduction about whether the trunk loads are used to assess instantaneous benefits associated with instantaneous intake rate or optimal foraging resource choice. A good number of references cited are about mechanistic relationships associated with instantaneous intake rate, whereas the use of trunk loads, or ‘bite mass’ for other herbivores, is in fact used as a proxy for intake rate and subsequently applied to seasonal foraging decisions. Trunk loads are average per resource types and habitats, not during continuous feeding bouts to approach intake rate, so the trade-off between bite size, chewing and processing, is not examine here. The intake rate model itself is dealt with in another paper by the same authors, so the focus is on trunk load as an index of foraging currency behind dietary choices and seasonal switches. May be this can be better expressed. May be the introduction could be streamlined by doing so.

Validity of the findings

The results are informative and contribute to a better understanding for why elephant are grass eaters in the wet season. My main concern is that I find some of the differences discussed and shown in the Figures 9 and 10 very small, and not very convincing. Because most the conclusions are based on the ‘bite mass’ component of the intake rate, I would not conclude to firmly on the choices based on intake rate, when handling and processes may also play significant roles in regulating intake rate. That said, the results give a convincing illustration that when green grass is available, optimal foraging does predict elephant should be grazers and they are. Data on isotopes and tooth wear show that indeed African elephants can shift towards grass eating when available, although remaining mixed feeders (Saarinen et al. 2015). Because tooth wear is very much associated with grass silicate content, especially in semi-arid environment, there may be several selective forces maintaining the mixed diet beyond short term intake rate!

Ref cited: Saarinen, J., Karme, A., Cerling, T., Uno, K., Säilä, L., Kasiki, S., … Fortelius, M. (2015). A New Tooth Wear–Based Dietary Analysis Method for Proboscidea (Mammalia). Journal of Vertebrate Paleontology, 35(3). https://doi.org/10.1080/02724634.2014.918546

---

## Round 0.2 · accepted · Accept

Thank you for addressing the reviewer's comments.

·

Basic reporting

No comment

Experimental design

No comment

Validity of the findings

No comment

Additional comments

Well done to the authors on a great paper! I look forward to seeing it published.